# The Feasibility and Accuracy of Holographic Navigation with Laser Crosshair Simulator Registration on a Mixed-Reality Display

**DOI:** 10.3390/s24030896

**Published:** 2024-01-30

**Authors:** Ziyu Qi, Haitao Jin, Qun Wang, Zhichao Gan, Ruochu Xiong, Shiyu Zhang, Minghang Liu, Jingyue Wang, Xinyu Ding, Xiaolei Chen, Jiashu Zhang, Christopher Nimsky, Miriam H. A. Bopp

**Affiliations:** 1Department of Neurosurgery, University of Marburg, Baldingerstrasse, 35043 Marburg, Germany; christopher.nimsky@uk-gm.de; 2Department of Neurosurgery, First Medical Center of Chinese PLA General Hospital, Beijing 100853, China; 18031275391@163.com (H.J.); wfwangqun@163.com (Q.W.); 15620946575@163.com (Z.G.); sy983246393@gmail.com (S.Z.); liumhdoctor@163.com (M.L.); richardwang9410@163.com (J.W.); thomascage@sina.com (X.D.); chxlei@mail.sysu.edu.cn (X.C.); shujiazhang@126.com (J.Z.); 3Medical School of Chinese PLA, Beijing 100853, China; 4NCO School, Army Medical University, Shijiazhuang 050081, China; 5Department of Neurosurgery, Division of Medicine, Graduate School of Medical Sciences, Kanazawa University, Takara-machi 13-1, Kanazawa 920-8641, Japan; xiongruochu@stu.kanazawa-u.ac.jp; 6Center for Mind, Brain and Behavior (CMBB), 35043 Marburg, Germany

**Keywords:** mixed reality, augmented reality, neuronavigation, laser crosshair simulator, neurosurgical planning, automatic registration, target registration error, dice similarity coefficient, head phantom, intracranial lesion

## Abstract

Addressing conventional neurosurgical navigation systems’ high costs and complexity, this study explores the feasibility and accuracy of a simplified, cost-effective mixed reality navigation (MRN) system based on a laser crosshair simulator (LCS). A new automatic registration method was developed, featuring coplanar laser emitters and a recognizable target pattern. The workflow was integrated into Microsoft’s HoloLens-2 for practical application. The study assessed the system’s precision by utilizing life-sized 3D-printed head phantoms based on computed tomography (CT) or magnetic resonance imaging (MRI) data from 19 patients (female/male: 7/12, average age: 54.4 ± 18.5 years) with intracranial lesions. Six to seven CT/MRI-visible scalp markers were used as reference points per case. The LCS-MRN’s accuracy was evaluated through landmark-based and lesion-based analyses, using metrics such as target registration error (TRE) and Dice similarity coefficient (DSC). The system demonstrated immersive capabilities for observing intracranial structures across all cases. Analysis of 124 landmarks showed a TRE of 3.0 ± 0.5 mm, consistent across various surgical positions. The DSC of 0.83 ± 0.12 correlated significantly with lesion volume (Spearman rho = 0.813, *p* < 0.001). Therefore, the LCS-MRN system is a viable tool for neurosurgical planning, highlighting its low user dependency, cost-efficiency, and accuracy, with prospects for future clinical application enhancements.

## 1. Introduction

In neurosurgery, commercial navigation systems have emerged as a standard paradigm, instrumental in aligning medical images with the physical patient space [1,2,3]. They enable neurosurgeons to accurately track surgical instruments and maximize the extent of resection while safeguarding critical anatomical structures during interventions [4,5,6]. These systems have employed landmark-based and surface-based methods alongside automatic registration methods, each with unique operational principles [1,2,4,5,6] (see Figure 1A,B). While automatic methods are lauded for their low user dependency input and reduced registration error, they also show some drawbacks. These especially include their substantial hardware requirements and the necessity of specialized operation rooms, factors that significantly elevate the cost and limit their widespread adoption [7,8].

In response to these challenges, there has been a shift towards exploring cost-effective alternatives, particularly in augmented reality (AR) and mixed reality (MR) technologies [9,10,11,12,13,14,15,16,17,18,19,20]. These technologies, particularly when implemented through a head-mounted display (HMD), offer an innovative approach for navigation. They provide an immersive, interactive experience, allowing for intuitive preoperative planning and surgical guidance, a concept called mixed reality navigation (MRN) [7,8,11,12,21,22,23]. Unlike conventional systems, MRN incorporates user-friendly interfaces and leverages emerging technologies to enhance accuracy and adaptability in surgical procedures [12,21,22,24,25,26].

The development of the laser crosshair simulator (LCS) within the MRN framework marks a significant advancement [7]. The LCS-MRN system, mirroring the automatic registration paradigm of conventional systems, aims to (1) reproduce the relationship between the reference scan and the physical patient, (2) establish an initial virtual-to-physical space registration, and (3) maintain this alignment through continuous tracking (see Figure 1C). The LCS-MRN system offers simplicity in hardware and software configurations, addressing the complexities of handling virtual objects.

Despite these advantages, two critical aspects arise regarding the LCS-MRN system’s reliability and clinical applicability. The first involves identifying a universal and reliable methodology to validate the accuracy of this novel system. The second aspect pertains to the system’s versatility and ability to meet the diverse and extensive demands of various surgical procedures and conditions. While various parameters have been studied within the MRN context, specific metrics to evaluate the LCS-MRN system remain unreported. Furthermore, its application in neurosurgery has not been extensively explored.

Therefore, this study retrospectively utilized imaging data from patients with intracranial lesions to create life-sized head phantoms for evaluation. These phantoms, employed in simulated clinical scenarios, facilitated the validation of the LCS-MRN system’s feasibility and accuracy. This approach attempts to address the gap in the literature by providing a foundational assessment of this innovative system in a controlled yet clinically relevant setting. Further, it presents an advancement in MRN registration techniques through a more streamlined and user-friendly hardware and software configuration. This enhanced approach significantly improves accuracy and flexibility and contributes to developing cost-effective MRN systems in neurosurgery. The potential for improved surgical outcomes, especially in environments without standard navigation systems, marks a significant advancement in neurosurgical technology.

## 2. Materials and Methods

This study aims to develop and validate a low-cost, easy-to-use LCS-MRN system for neurosurgical planning and navigation. It addresses the dual objectives of creating this innovative system and establishing a universal, reliable evaluation methodology for its accuracy and applicability in diverse neurosurgical contexts. This section provides a detailed overview of the foundational concepts and structure of the LCS-MRN system, followed by an exploration of the techniques and methods used for image processing and phantom creation. The section concludes by detailing the metrics used to evaluate the quality of the LCS-MRN system. The schematic of the practical workflow is depicted in Figure 2.

### 2.1. Development of the LCS-MRN System

#### 2.1.1. The Laser Crosshair Simulator (LCS)

The LCS, as depicted in Figure 3, was designed to address the challenges of scanner tracking in MRN. Conventional automatic registration systems often rely on a reference array attached to the scanner, aligning images with a universal coordinate framework such as the world coordinate system [7]. However, the LCS introduces a novel approach by employing dual laser emitters and a specialized MR interface to overcome disruptions in optical tracking caused by user movement or scanner activities in HMDs (see Figure 3B). These lasers project intersecting lines, simulating the crosshairs found in computed tomography (CT) or magnetic resonance imaging (MRI) scans, facilitating the precise transfer of scanning details for accurate image localization (see Figure 3A). The MR interface features a stainless-steel panel with a target image on both sides. When recognized by the HMD’s Vuforia software development kit (SDK) (Version 10.14, PTC, Inc., Boston, MA, USA), the image’s central point establishes the reference origin in virtual space, ensuring precise hologram positioning and enhancing spatial mapping accuracy in MRN systems (see Figure 3C).

#### 2.1.2. The Mixed Reality Navigation (MRN)

The MRN system in this study utilizes the Microsoft HoloLens-2 (HL-2) (Microsoft, Redmond, WA, USA) as its primary hardware. The HL-2, a self-contained, portable, optical lightweight HMD, operates independently without external hosts or tracking systems. Equipped with a high-definition red–green–blue (RGB) camera, four Visible Light Cameras (VLC), a depth camera, and an Inertial Measurement Unit (IMU), the HL-2 excels in mapping its position and orientation through simultaneous localization and mapping (SLAM). It projects virtual objects onto a specific plane using waveguide imaging technology, offering a 43° × 29° field of view (FoV) color display, which ensures stable, realistic virtual representations regardless of the viewer’s perspective.

The software component of the MRN system is developed in Unity (Version 2021.3.4f1, Unity, San Francisco, CA, USA), integrated with image detection and tracking algorithms from the Vuforia SDK (Version 10.14, PTC, Inc., Boston, MA, USA). The interface, designed using the Mixed Reality Toolkit (MRTK) SDK (Version 2.8.3, Microsoft), employs C# scripts in Visual Studio (Version 16.11.26, Microsoft, 2019) for program logic and user interaction through voice and gesture commands. This setup facilitates an immersive experience, with the final application packaged as a Universal Windows Platform (UWP) app tailored for deployment on the HL-2. The combination of these advanced software elements enhances the overall functionality and user experience of the MRN system.

### 2.2. Image Data Acquisition and Image Processing

#### 2.2.1. Subjects

The study retrospectively collected preoperative cranial MRI or CT data from 19 patients over four years (2018–2021) in two clinical hospitals (the first medical center of the Chinese PLA General Hospital, Beijing, and the Hainan Hospital of the Chinese PLA General Hospital, Sanya) who were diagnosed with intracranial lesions (neurological neoplasm, hypertensive cerebral hemorrhage) and were candidates for surgical intervention. Prior to study inclusion, written informed consent was provided by each patient or their legal representative, permitting the use of pseudonymized image data for this study. Given the study’s retrospective nature and the absence of invasive procedures, an ethical review was deemed unnecessary. The 19 enrolled patients met the following minimal inclusion criteria:High-resolution and high-quality MRI or CT data with clear skin contours covering the cranial region;Clear lesion boundaries visible in at least one image data set;At least five skin adhesive markers were attached to the patient’s scalp before imaging.

In this study, all patients underwent surgery guided by conventional navigation systems, where the markers placed near the surgical area were used for traditional navigation registration [8,23]. The successful completion of these surgeries without significant complications validated the effectiveness of the neuroimaging and skin markers used for traditional navigation. This outcome is crucial, as it establishes the reliability of the preoperative data and marker placements, which are also essential for evaluating the quality of the MRN system. By confirming that these elements were effective in a conventional setting, one can reasonably infer their utility in evaluating the MRN system. Thus, the surgical procedures and their outcomes, although not directly involving MRN, provide a foundational basis for the MRN study, ensuring that the retrospective data used is accurate and applicable for this novel navigation system evaluation.

#### 2.2.2. Image Acquisition

In the case of intracranial neoplasms, high-resolution image acquisition was conducted using a 1.5 T MRI scanner (Espree, Siemens, Erlangen, Germany), whereas, in the case of cerebral hemorrhage, image acquisition was performed using a 128 multislice CT scanner (SOMATOM, Siemens, Forchheim, Germany). All image data were acquired one day before or on the day of the surgery, serving as the reference image for reconstructing the medical imaging space’s geometric shape, precisely the scanning parameters the LCS-MRN system aimed to replicate and transfer in this study. They were thus termed “reference scans”, with the resulting images called “reference images”. In addition to the reference images, all patients with intracranial neoplasms underwent multimodal MRI acquisition days before surgery, including, depending on the location of the lesion, a T1-weighted contrast-enhanced (T1-CE), T2-weighted (T2) and/or Diffusion tensor imaging (DTI)) in clinical routine. Imaging data were provided in Digital Imaging and Communications in Medicine (DICOM) file format.

#### 2.2.3. Creation of Head Phantoms

Life-sized head phantoms for all 19 subjects were 3D printed based on the individual MRI or CT-based reference image to evaluate the practicality of the suggested registration technique. Firstly, standard tessellation language (STL) files, a widely used 3D printing and modeling format, were generated from segmented skin surfaces in reference imaging using the “Segment Editor” extension within 3D Slicer software (Version 5.1.0) for 3D reconstruction. The process also involved model optimization, such as hollowing the head, smoothing the surface to eliminate noise, ensuring a flat bottom surface for stable printing, and refining sharp edges for the researcher’s safety to ensure high-quality, cost-effective, and safe 3D phantom creation. Utilizing the basic planes from the reference image, the laser crosshair mark lines (width: 1 mm) were reconstructed and incorporated into the design of the phantom (see Figure 4B). The replica was exported as an “.stl” file and manufactured via an A5S 3D printer (Shenzhen Aurora Technology Co., Ltd., Shenzhen, China) under the following conditions: extrusion temperature 210°C, bed temperature 50 °C, polylactic acid as print medium, layer height 0.3 mm, and infill density 10%, thereby producing a life-sized head phantom for assessment.

#### 2.2.4. Image Segmentation

To rigorously evaluate the MRN system’s accuracy and reliability, a set of outlined objects was generated to create holograms using the 3D Slicer platform for this investigation (see Figure 4A). For each subject, intracranial lesions (neurological neoplasm or hematoma) were segmented following a strict protocol to evaluate the MRN system’s precision in lesion localization. Additionally, fiducial markers were segmented, and their centroids were marked to elaborate a series of accuracy metrics quantitatively (as indicated in Section 2.4.2). Furthermore, for each case, critical anatomical structures (such as ventricles, arteries, venous sinuses, pyramidal tracts, optic radiations, frontal sinuses, and scalp quadrants) upon availability of according data sets, as well as models for surgical guidance (such as puncture paths for reaching hemorrhage, endoscopic pathways, and the bone flaps suitable for port surgery with introducer assistance), were individually segmented [7,8,23,27], allowing to simulate the visualization environment in clinical settings closely and to evaluate the MRN system’s computational load capacity for rendering multiple models. It should be noted, however, that due to individual variations and different surgical plans, these structures were not used for accuracy assessment.

Markers and scalp quadrants were segmented within the T1-weighted data set. The segmentation of intracranial lesions, arteries, venous sinuses, and ventricles was carried out based on the T1-CE data. In contrast, the identification and segmentation of the optic radiations and pyramidal tracts were based on the DTI data. The latter utilized a specialized and distinctly different approach from conventional image segmentation, known as tractography [28,29,30].

Finally, co-registration was performed across all image data sets to transform all segmented structures from multiple modalities into a unified reference image coordinate system (RICS). Therefore, the reference images (i.e., the highest-resolution T1-weighted scan of the cranium with attached markers or the CT data set, respectively) were set as fixed images while all other data sets (T1-CE, T2, and DTI) were rigidly co-registered to this reference image. The registration process was facilitated through the “General Registration (Elastix)” extension available on the 3D Slicer platform, with the preset set to “Generic rigid (all)”. The calculated registration matrix was saved in the 3D Slicer scene files, enabling the transformation of various segments into a unified coordinate system.

#### 2.2.5. Holograms Generation

Through the “Segmentations” extension on the 3D Slicer platform, pixels segmented on each two-dimensional (2D) slice were aligned, stacked, and voxelized in three-dimensional space to construct continuous 3D surfaces for holographic visualization (3D holograms). This process, known as 3D reconstruction, liberates the user’s observation of holograms from the confines of the original imaging planes. All models were exported in the OBJ file format.

### 2.3. LCS-MRN Registration Workflow

#### 2.3.1. LCS Deployment

The generated head models and holograms were used for further analyses (see Figure 4A,B). Phantoms were securely affixed using self-tapping screws from their base to prevent any unknown deformations during fixation that might affect the accuracy of measurements (see Figure 4C) as the patient’s head in a clinical setup would be fixed with a head clamp too during navigation-supported surgery. Once the head phantom was securely positioned, the user turned on the laser emitters and fine-tuned the placement of the LCS (see Figure 4D). This adjustment continued until the LCS’s laser lines were precisely aligned with the marked lines on the phantom. Subsequently, the LCS’s orientation was fixed to preserve its alignment relative to the phantom.

#### 2.3.2. Surgical Position Compatibility

The study developed a matrix transform approach encapsulated within the software to facilitate LCS deployment and practicality in varying surgical positions. Since reference imaging is usually conducted in the supine position, differing from potential prone or lateral surgical positions, using the original coordinates (i.e., the RICS) directly could lead to inconvenient LCS placement, possibly conflicting with surgical instruments or anesthesia equipment. For example, in a prone position, direct use of original coordinates might place the LCS in conflict with the patient’s head fixation system. Similarly, in a lateral position, the LCS could potentially interfere with the anesthesia tubing. The authors’ software-based matrix computation artificially adjusts the coordinate system—for instance, rotating the prone position around the vertical axis by 180° and the lateral position by ±90°. This allows for LCS placement that does not interfere with surgical operations while maintaining patient positioning and smooth medical procedures.

To ensure compatibility of the holograms with the LCS-MRN system, in addition to adapting for different surgical positions as mentioned earlier, it is also essential to address the disparities in scale between the Reference Image Coordinate System (RICS) and the Virtual Coordinate System (VCS) defined by the Vuforia SDK. For example, while RICS is usually defined in millimeters during imaging scans, the Vuforia SDK operates in centimeters, necessitating careful consideration of unit conversion and corresponding numerical transformations. Considering the differences in coordinate axis directions and coordinate scales, the Forward Engineering Matrix (FEM) is implemented to achieve compatibility with the Vuforia SDK. This step integrates both the direction and scale disparities to precisely position, orient, and dimension the virtual objects within the LCS-MRN system, ensuring accurate holographic visualization in varied surgical positions and setups (see Figure 5A–L). In other words, FEM harmonizes the physical and virtual environments for effective LCS deployment. Conversely, when analyzing holograms from the MRN system within 3D Slicer, involving the analysis coordinate system (ACS), a reciprocal operation equivalent to reversing the linear transformation, known as the Reverse Engineering Matrix (REM), becomes imperative. These transformations are contingent upon the patient’s head position during the use of the LCS, with various positions such as supine, prone, left-lateral, and right-lateral necessitating distinct FEMs and REMs. Notably, since RICS and ACS are right-handed coordinate systems while VCS is a left-handed system, a handedness conversion matrix (HCM) is required to invert the direction of the z-axis, a step encapsulated within the software; hence, in practice, the FEM is the transformation from RICS to an intermediate VCS (IVCS).

Before being integrated into the MR platform, all models were adapted using the FEM-based process via the “Transform” extension in the 3D Slicer software.

#### 2.3.3. MRN Setup

The HL-2 was carefully configured to each user’s interpupillary distance (IPD), ensuring a custom-fit visual experience to initialize the holographic visualization. The device must be securely positioned on the user to prevent accidental shifts. Utilizing Vuforia SDK’s feature detection algorithm, the MRN system meticulously tracks the designated image markers within the LCS, thus accurately maintaining the real-time positioning of the LCS. This precise tracking facilitates the establishment of the RICS defined by the LCS’s orientation and position. Subsequently, the user employs gesture controls to introduce the holographic model into the RICS. The holograms were visualized on the static head phantom, signifying the completion of the MRN’s initial registration phase (see Figure 4E and Figure 6A).

#### 2.3.4. Registration Update

Subsequently, users can maintain or modify the current registration. This is controlled by switching between the “Freeze” and “Unfreeze” modes on the MR platform. Activating “Freeze” secures the holographic display to its spatial anchor, achieved through the HL-2’s SLAM process, thus ensuring stability of the holographic image even during temporary tracking loss of the image target. Conversely, selecting “Unfreeze” resumes the tracking process, allowing for updates to the hologram to reflect any shifts in the patient’s head position relative to the global coordinate system.

### 2.4. Quality Assessments

#### 2.4.1. Practicality Assessment for the Workflow

The workflow and the visualized scenes of the LCS-MRN for each case were retrospectively analyzed to assess the system’s practicality in a simulated clinical setting. This evaluation was facilitated through the HL-2’s ’Mixed Reality Capture (MRC)’ feature, which allowed for capturing the merged virtual and physical worlds into photo or video files using the built-in Photos-Videos (PV) camera. To mitigate any potential impact on the device’s performance and spatial positioning accuracy, MRC was primarily conducted via photography. The recorded images were then systematically reviewed to evaluate the ease of use, accuracy of virtual object placement, and overall user interaction within the MR environment. These factors are crucial in determining the practicality of the LCS-MRN system in real-world clinical scenarios. Multiple methods were available to activate the camera, including local control through gaze, hand gestures, and voice commands or remote operation via a web browser on a connected computer or smartphone, offering flexibility in documentation.

#### 2.4.2. Assessment of Landmarks

The fiducial markers attached to the scalp, not included in the registration process, were selected as reference points for accurate measurement. In this study, the LCS-MRN system’s custom-developed “virtual probe” feature was utilized to acquire coordinates of the virtual (fiducial holograms) and the physical representation (3D-printed markers integrated with the head model) of these markers. The virtual probe, designed with a white line handle and a spherical tip, is a precise tool within the virtual space, enabling the user to position it accurately at spatial points of interest. As the user positions the probe tip at the chosen spatial locations, the LCS-MRN system relays the tip coordinates within the RICS on its virtual panel. It is imperative to note, as outlined in Section 2.2.5, that the coordinates collected for each case must transform the (REM) corresponding to the surgical position. This step is essential before proceeding to further analysis in 3D Slicer; failure to do so could lead to significant errors. The coordinates of both the virtual and physical fiducials, along with their centroid coordinates analyzed using 3D Slicer, are fundamental for calculating the system’s interpolative and extrapolative accuracy indices.

The marker’s centroids were considered the ground truth in the RICS with right–anterior–superior (RAS) coordinates (see Figure 6C), a standard medical coordinate system where the primary axes of right, anterior, and superior are considered positive directions. These centroids can be extracted using the “Segment statistics” module within 3D Slicer. For computational convenience, unique indices are established for each marker, allowing each case to have an ordered set of fiducial markers, C, facilitating rapid access and manipulation of specific points. Without loss of generality, the centroid coordinates of the *i*-th fiducial marker, $C_{i}$, are denoted as:(1)$$C_{i}={\left(x_{C_{i}},y_{C_{i}},z_{C_{i}}\right)}^{⊤}$$

For each marker, relying on stereoscopic vision, the user delicately maneuvered the virtual probe’s tip and sequentially placed it on each perceived virtual marker point, $V_{i}$ (see Figure 4E and Figure 6B). The MR platform instantaneously reported the converted coordinates of this point within the RICS in a panel as:(2)$$V_{i}={\left(x_{V_{i}},y_{V_{i}},z_{V_{i}}\right)}^{⊤}$$

Similarly to the point set C, all virtual marker points compose an ordered set, V, based on the indices of the markers. The presented coordinates in the panel can be exported for subsequent analysis.

Following this, the users cleared all probes and repositioned the virtual probe’s tip sequentially on each perceived physical marker point, $P_{i}$ (see Figure 4G and Figure 6B). The MR platform also reported and exported $P_{i}$’s coordinates within the RICS, formulating another ordered set, P, expressed as:(3)$$P_{i}={\left(x_{P_{i}},y_{P_{i}},z_{P_{i}}\right)}^{⊤}$$

#### 2.4.3. Landmark-Based Evaluation of Accuracy

To objectively quantify the discrepancy between virtual images and the actual physical environment in the LCS-MRN system, the target registration error (TRE) was chosen as the measure of accuracy. The TRE is calculated based on the Euclidean distance between specific reference points in virtual space and their corresponding points in physical space. However, considering potential manual errors that might occur during the use of the virtual probe, the fiducial localization error (FLE) was additionally utilized as an indicator of the reliability of the assessment process. Furthermore, the fiducial registration error (FRE) and the Frobenius norm (FN) were analyzed as extrapolative indicators of overall head displacement, as the anatomical accuracy of the patient’s head itself, compared to the fiducials.

The FLE denotes the displacement vector from a marker’s centroid (ground truth) to the perceived virtual counterpart’s center, reflecting the perceptual error as the user moves and positions the virtual probe’s tip. A minimal FLE indicates a trustworthy spatial point measurement by the user. FLE at the *i*-th fiducial marker can thereby be computed as:(4)$${\mathrm{FLE}}_{i}=V_{i}-C_{i}$$

Its magnitude can be computed as: (5)$$∥{\mathrm{FLE}}_{i}{∥}_{2}=\sqrt{{\left(x_{V_{i}}-x_{C_{i}}\right)}^{2}+{\left(y_{V_{i}}-y_{C_{i}}\right)}^{2}+{\left(z_{V_{i}}-z_{C_{i}}\right)}^{2}}$$

The term ${∥\cdot ∥}_{2}$ denotes the Euclidean norm of the displacement vector.

The TRE represents the displacement vector post-registration from a marker’s centroid (ground truth) to the perceived physical counterpart’s center, signifying the virtual-to-physical alignment accuracy. It acts as a validity metric for the data set. A lower magnitude of TRE insinuates accurate virtual–physical alignment by the MRN system. It can be formulated as follows:(6)$${\mathrm{TRE}}_{i}=P_{i}-C_{i}$$

Its magnitude can be computed as:(7)$$∥{\mathrm{TRE}}_{i}{∥}_{2}=\sqrt{{\left(x_{P_{i}}-x_{C_{i}}\right)}^{2}+{\left(y_{P_{i}}-y_{C_{i}}\right)}^{2}+{\left(z_{P_{i}}-z_{C_{i}}\right)}^{2}}$$

The FRE describes the displacement vector between two registered ordered point sets. Specifically, when the markers’ centroid (as the ground truth, GT) set C is transformed to align the perceived physical point sets P using an algorithm (e.g., least squares), FRE measures the displacement vector between each point in P and its counterpart point in the transformed set $P^{*}$ (see Figure 4H and Figure 6D–F) and gauges the congruence between the two sets of points. A reduced FRE indicates a high level of geometric shape consistency between the virtual and physical point sets, thereby suggesting the reliability of extrapolative analysis based on reference landmarks. The FRE at the *i*-th marker can be formulated as:(8)$${\mathrm{FRE}}_{i}=P_{i}^{*}-P_{i}$$

Given that the point sets C and P contain an equal number of points with corresponding indices, the optimal rigid linear transformation mapping from C to P is appropriately computed using a least squares algorithm. Calculation of the optimal transformation can be automatically performed in the “Image-guided therapy (IGT)” module of the 3D Slicer platform in the “Fiducial Registration Wizard” extension. The core algorithm of this extension, an overwriting of the “vtkLandmarkTransform()” function in the Visualization Toolkit (VTK) library, accomplishes the task of fitting the optimal rigid linear transformation between two ordered point sets, subsequently returning the optimal transformation matrix $T_{C}^{P*}\in R^{4\times 4}$ to the user (see Figure 4I and Figure 6F). $T_{C}^{P*}$ can be represented as:(9)$$T_{C}^{P*}=\left[\begin{array}{cc} R_{C}^{P*} & t_{C}^{P*}\\ 0 & 1 \end{array}\right]$$

The term $R_{C}^{P*}\in R^{3\times 3}$ is the rotation matrix, $t_{C}^{P*}\in R^{3\times 1}$ is the translation matrix, and $0\in R^{1\times 3}$ represents no affine transformation on the original points. One point $P_{i}^{*}$ in the post-transformation point set P can be expressed as:(10)$$P_{i}^{*}=R_{C}^{P*}C_{i}+t_{C}^{P*}$$

Based on the definition of FRE, the following relationship is derived:(11)$${\mathrm{FRE}}_{i}=P_{i}^{*}-P_{i}=R_{C}^{P*}C_{i}-P_{i}+t_{C}^{P*}$$

Incorporating the coordinates $P_{i}$ and $C_{i}$ into Equation (Equation 11) facilitates the calculation of the components of ${\mathrm{FRE}}_{i}$ along the three principal axes in the RICS and its Euclidean norm $∥{\mathrm{FRE}}_{i}{∥}_{2}$.

The FN offers a method for quantifying the “size” of a displacement matrix, analogous to the Euclidean vector norm but applicable to matrices. In the context of precision assessment, the FN can be used to measure the discrepancy between matrix $T_{C}^{P*}$ and the identity matrix $I\in R^{4\times 4}$ across all elements. A larger FN of the difference between $T_{C}^{P*}$ and I indicates a greater divergence between $T_{C}^{P*}$ and I, signifying a higher mismatch between virtual elements and their physical space counterparts following MRN registration. The FN of the difference between $T_{C}^{P*}$ and I is calculated using the following formula:(12)$$FN=∥T_{C}^{P*}{-I∥}_{F}=\sqrt{\sum_{i=1}^{4}\sum_{j=1}^{4}{\left|{\left(T_{C}^{P*}-I\right)}_{ij}\right|}^{2}}$$

#### 2.4.4. Lesion-Based Evaluation of Accuracy

Precise lesion localization, encompassing both volume and boundaries, is essential for successful surgical planning and navigation. In this study, the Sørensen–Dice similarity coefficient (DSC) and the 95th percentile of the Hausdorff distance (HD_95_) were chosen to analyze the LCS-MRN system’s performance, focusing on lesion volume and shape accuracy. To assess the accuracy of the alignment between virtual and physical lesions, the ground truth (**GT**) and the transformed model (**TM**) of the lesion(s) in the physical scene were assessed using both volumetric and surface-based metrics. In this study, the TM is approximated by the lesion segmentation results upon which the transformation $T_{C}^{P*}$ is applied. The “Segment compare” extension in 3D Slicer facilitates the automatic computation of the DSC and HD_95_.

For analysis regarding the spatial volumetric overlap, the DSC is calculated as follows: (13)$$DSC=\frac{2|\mathrm{GT}\cap \mathrm{TM}|}{\left|\mathrm{GT}|+|\mathrm{TM}\right|}$$

Here, **TM** denotes the point set of the transformed model, and **GT** is the point set representing the ground truth. This metric ranges from 0 to 1, whereas a value of 1 indicates perfect congruence, while 0 denotes no spatial overlap.

To analyze the object’s shape similarity, the HD_95_ is used, calculating the maximum distance between corresponding points on the surface point sets, mitigating the effect of outliers by focusing on the 95th percentile range of the measurements. This distance is reflected in millimeters and is expressed through the formula: (14)$$HD_{95}=\max_{gt\in \mathrm{GT}}^{95\%}\left(\min_{tm\in \mathrm{TM}}{∥tm-gt∥}_{2}\right)$$

In this formula, $\max_{gt\in \mathrm{GT}}^{95\%}\left(\cdot \right)$. Calculate the maximum value of the first 95 % of the distance set, where TM denotes the set of contour points from the transformed model, and GT corresponds to the set of contour points from the ground truth.

To investigate whether these accuracy metrics related to the lesions correlate with the lesions’ intrinsic characteristics or initial scanning positions; in this study, the lesion volume, the lesion depth, and L2 norm are calculated for each lesion as follows:The **lesion volume** is defined as the product of the number of voxels contained in the lesion’s segmentation and the physical volume of a single voxel, calculated using the “Segment statistics” module in the 3D Slicer software (Version 5.1.0).The **lesion depth** is defined as the minimum radius of a sphere tangential to the skin surface with the lesion’s centroid as its center. Therefore, the centroid of the lesion (geometric center) is first calculated using the “Segment statistics” module. Subsequently, the radius of this tangential sphere is computed using the “Model to model distance” module in the 3D Slicer software (Version 5.1.0).The **L2 norm** is defined as the Euclidean distance from the lesion’s centroid to the origin of the reference image space in the 3D Slicer software (Version 5.1.0).

#### 2.4.5. Statistical Analysis

Statistical analyses were conducted using the Kruskal–Wallis test across all 19 cases for landmark-based metrics and all 21 lesions for volumetric assessments. The Mann–Whitney U-test was utilized further to investigate differences between groups in our post hoc analysis. The overall level of significance was set to *p* < 0.05. Considering multiple comparisons being performed (three different surgical positions), the Bonferroni correction was applied to adjust the alpha value to control for the overall Type I error rate (*p* < 0.017 (0.050/3)) to control for the overall Type I error rate in these comparisons. Correlations were analyzed separately for DSC and HD_95_ with lesion and scanning characteristics (volume, depth, and L2 norm) using Spearman’s rank correlation method. MATLAB (version R2022a, MathWorks, Apple Hill Campus, Natick, MA, USA) was utilized for all statistical computations and the generation of plots.

## 3. Results

This section encompasses a comprehensive quality evaluation of the LCS-MRN system. Section 3.1 details patient demographics and clinical profiles involved in the study. Section 3.2 discusses hologram generation and application for anatomical visualization to assess the system’s practicality. Section 3.3 focuses on a landmark-based accuracy evaluation, quantifying the virtual–physical discrepancies. Finally, Section 3.4 emphasizes lesion-based analysis, assessing the system’s precision in lesion localization for navigation, underlining the LCS-MRN system’s clinical utility.

### 3.1. Subject Demographics

In this cohort of 19 patients, comprising twelve males and seven females with an average age of 54.4 ± 18.5 years, 15 patients underwent preoperative MRI imaging in case of an intracerebral neoplasm (case 01 to 15), while four underwent CT imaging (case 16 to 20) in case of intracerebral hematoma. The patient’s demographic information, histopathological findings, and surgical positioning are shown in Table 1. Intracranial neoplasms included metastasis (n = 7), meningioma (n = 3), diffusion astrocytoma (n = 1), cavernous malformation (n = 1), diffuse large B-cell lymphoma (n = 1), aneurysmal alterations (n = 1) and high-grade glioma (n = 1). In total, 21 lesions were identified, with four lesions in the frontal, three in the parietal, two in the temporal and seven in the occipital lobe. Additionally, three lesions were located in the basal ganglia, and two were infratentorial. Furthermore, hematoma is identified in four subjects.

All their life-sized head phantoms with positioning lines were successfully created. Based on the surgical records and photo documentation, the surgical positioning of all head phantoms was accurately replicated. Eight phantoms were secured in a supine position, six in a prone position, and five in a lateral position, with three left-sided and two right-sided.

### 3.2. Practicality Assessment Results

A comprehensive set of holograms (n = 240) was generated for experimental purposes (see Figure 7 and Table 2). Specifically, the overall count of holograms created includes 21 instances of lesions, 15 pairs of arteries and venous sinuses, 19 ventricles, seven pairs of the optic radiation and pyramidal tract, four puncture paths, three frontal sinuses, and three pairs of endoscopic pathways and bone flaps. Additionally, 19 scalp quadrants serving as intuitive references for alignment evaluation and 124 markers serving as reference points for quantitative accuracy assessment were visualized.

For all 19 cases, 240 holograms were visualized successfully using MRN (see Figure 8). It is noteworthy that the image targets on the LCS’s MR interface were affixed on both sides, ensuring that regardless of the LCS being positioned on the user’s left (Cases 10, 11, 14) or right side (all other cases), the image targets were consistently recognized and tracked by the HL-2 system, yielding successful registration and visualization (see Figure 9).

The LCS-MRN approach effectively facilitated comprehensive intracranial anatomical visualization. The LCS deployment took 2 to 3 min, and the hologram loaded onto the MR platform for approximately 1 to 2 min.

### 3.3. Landmark-Based Evaluation of Accuracy

The landmark-based analysis compares the LCS-MRN system’s interpolated metrics (FLE and TRE) and extrapolated metrics (FRE and FN) across various surgical positions, including supine, prone, and lateral. Post hoc analysis was also conducted to assess statistical significance between different position groups.

In the case-oriented analysis, for interpolated metrics for accuracy, the FLE across all cases was 1.9 ± 0.4 mm, and the TRE was 3.0 ± 0.5 mm (see Figure 8 and Figure 10 and Table 3). The Kruskal–Wallis test revealed no statistically significant differences in FLE and TRE across different surgical positions (supine, prone, and lateral) (FLE 2.0 ± 0.4 mm vs. 1.8 ± 0.2 mm vs. 1.8 ± 0.7 mm, *p* = 0.628; TRE 2.9 ± 0.6 mm vs. 3.1 ± 0.6 mm vs. 3.0 ± 0.4 mm, *p* = 0.745). For extrapolated metrics for accuracy, FRE for all cases was 2.1 ± 0.6 mm, and the FN was 3.4 ± 1.7 (see Figure 8 and Table 3). The Kruskal–Wallis test indicated no significant differences in FRE across surgical positions (FRE 1.9 ± 0.6 mm vs. 2.4 ± 0.5 mm vs. 2.2 ± 0.4 mm, *p* = 0.154); however, significant differences were observed in the FN between groups (2.3 ± 0.7 vs. 4.2 ± 1.5 vs. 4.0 ± 2.2, *p* = 0.034). In post hoc analysis using the Mann–Whitney U-test, the alpha value was Bonferroni-adjusted to 0.017 (0.050/3) to control for the overall Type I error rate in multiple comparisons, but no pairwise comparison showed statistically significant differences (supine vs. prone, *p* = 0.020; supine vs. lateral, *p* = 0.653; prone vs. lateral, *p* = 0.662).

### 3.4. Lesion-Based Evaluation of Accuracy

The lesion-based analysis of the LCS-MRN system evaluates metrics such as DSC and HD_95_ alongside lesion characteristics, including lesion volume, lesion depth, and L2 norm. Statistical correlation analyses and post hoc tests were included to explore their impact on the LCS-MRN system’s performance.

In the lesion-oriented analysis, the volume for all 21 lesions was 23.9 ± 26.3 cm^3^, lesion depth was 3.8 ± 1.5 cm, L2 norm was 8.0 ± 2.3 cm, DSC was 0.83 ± 0.12, and HD_95_ was 0.76 ± 0.15 mm (see Figure 8 and Table 4). The Kruskal–Wallis test indicated statistically significant differences in DSC and HD_95_ among lesions in different surgical positions (DSC 0.90 ± 0.06 vs. 0.76 ± 0.15 vs. 0.81 ± 0.09, *p* = 0.005; HD_95_ 1.7 ± 0.8 mm vs. 3.2 ± 0.6 mm vs. 1.9 ± 0.7 mm, *p* = 0.042). In post hoc analysis, the alpha value was Bonferroni-adjusted to 0.017 (0.050 / 3) to control for the overall Type I error rate for multiple comparisons. Post hoc analysis for HD_95_ revealed significant differences between prone and supine (*p* = 0.002) and between prone and lateral (*p* = 0.009), but no significant difference between supine and lateral positioning (*p* = 0.489). However, post hoc analysis using the Mann–Whitney U-test for DSC did not show statistically significant differences in pairwise comparisons (supine vs. prone, *p* = 0.029; supine vs. lateral, *p* = 0.059; prone vs. lateral, *p* = 0.534).

Additionally, the Kruskal–Wallis test demonstrated statistically significant differences in lesion depth and L2 norm across lesions in different surgical positions (depth, *p* = 0.012; L2 norm, *p* = 0.005) (see Figure 8 and Table 4). Although the variance in volume across different surgical positions was not statistically significant (*p* = 0.069), it is noteworthy that the mean values for the supine and prone groups were more than double that of the lateral group, indicating that lesion and scanning attributes were not evenly distributed among the surgical position groups. Therefore, these results do not imply that variations in surgical positions affect the magnitude of DSC and HD_95_.

The correlation analysis for DSC and HD_95_ with characteristics (lesion volume, lesion depth, and L2 norm), conducted using Spearman’s rank correlation method, revealed that apart from a statistically significant positive correlation between volume and DSC (rho = 0.813, *p* < 0.001), no other characteristics exhibited statistically significant relationships (see Table 5). When all characteristics were divided into groups based on greater than or less than the median value, only the volume size significantly impacted DSC (0.91 ± 0.04 vs. 0.76 ± 0.11, *p* < 0.001). The other groups’ differences were not significant (see Table 6). These results are consistent with the correlation analysis findings.

Figure 8 displays the extrapolated displacement and visualization results of the entire head and intracranial structures.

## 4. Discussion

Central to the study is evaluating an innovative MRN registration approach using an LCS, which addresses challenges such as user dependency and being cost-effective while enhancing accuracy in neurosurgical procedures. The method effectively replicates the scanner frame’s position relative to the patient and autonomously performs transformations, aligning tracking space coordinates with the imaging space. This study retrospectively utilized image data from a cohort of patients with intracranial lesions to fabricate life-sized head phantoms. These were then employed in simulated clinical environments to test the LCS-MRN system, focusing on registering image data and head phantoms to ascertain the system’s effectiveness and accuracy. The results demonstrated encouraging outcomes, laying the groundwork for future refinements and clinical applications.

The evolution of MRN can be traced through a series of advancements and challenges. Initial stages featured user-directed adjustments of virtual elements to correspond with real anatomical structures, a method necessitated recurrent recalibrations due to positional changes, thus being inefficient and inconsistent [11,12]. Advancements led to adopting a fiducial-based registration method, using distinct markers for more rapid configuration than the manual approach, albeit at the cost of increased FLE from marker displacement (e.g., skin shift) or equipment degradation (e.g., pointer abrasion) [8,21,23,31]. Meanwhile, some evolution shifted towards a markerless, surface-oriented strategy deploying computer vision (CV) algorithms, automating patient-image congruence, and minimizing physical interaction [32,33,34,35]. While streamlining the procedure during this phase, issues were encountered in maintaining registration accuracy due to sensitivity to low image quality, surface irregularities, feature scarcity, and image distortion, all demanding high computational power.

Amidst these developments, the quest for more efficient MRN methods persisted. The LCS-based registration method streamlines MRN by simplifying user involvement by aligning two laser cross-projections, surpassing the complexities of virtual object manipulation. This technique bypasses issues such as pointer degradation and the necessity for pre-defining virtual registration aids, as it employs pre-scanned, user-independent reference planes from DICOM images [7]. These orthogonally arranged planes ensure a consistent and globally representative registration process. The system is also user-friendly in assembly and operation, characterized by low-cost hardware fabrication and software processes that demand minimal computational power. A comparative analysis of registration techniques within state-of-the-art MRN paradigms is presented in Table 7, which elucidates the relative advantages and limitations of the LCS-MRN system against conventional systems.

This study’s integration of the LCS with the MRN system presents several navigational benefits. Technically, the LCS offers surgeons an intuitive physical positioning guide, complemented by the MR platform’s detailed visualization of anatomical structures, enhancing surgical area understanding. Visually, the LCS provides stable tracking in the physical space, while the MR platform maintains holographic consistency, even when surgeons adjust their perspectives during the procedure. The swift deployment of the LCS during surgery preparation, coupled with the MR platform’s visual aids, streamlines the surgical workflow. This “hybrid combination” approach, based on the principles of complementarity and compatibility, ensures that each system’s strengths are utilized while compensating for any limitations. This novel application of the LCS in establishing a consistent coordinate origin with CT or MRI data marks a significant advancement in MRN, offering improved integration and alignment accuracy. Bridging these technical and visual advantages, the dual-sided capability of the LCS’s MR interface marks a strategic innovation that optimizes spatial utility in the operating room (OR). This adaptability ensures the integration system’s functionality is maximized, allowing for flexible positioning that accommodates the surgeon’s needs and the constraints of the surgical environment.

Building on the advancements introduced by the LCS-MRN system, its versatility and scalability further extend its potential impact in the medical field. The scalability of the LCS-MRN system is highlighted by its effectiveness in various surgical positions, overcoming the limitations of traditional MRN paradigms, which are often restricted to supine positioning. The proposed system is equally effective in prone and lateral positions, broadening its applicability in diverse surgical contexts. Moreover, its foundational design centered on reference scans and integration with RICS presents the potential for compatibility with diverse imaging modalities, potentially including ultrasound and functional MRI. As long as modalities can align with RICS, they can integrate seamlessly with the LCS-MRN system, extending its applicability. Furthermore, the streamlined structural design of the LCS-MRN allows for adaptation across varied clinical settings, from advanced operating rooms to resource-limited environments, augmenting its global applicability and addressing a spectrum of medical requirements.

As a novel MRN approach, LCS-MRN aims to enhance neurosurgeons’ visual and spatial skills through perception and interaction. Theoretically, procedures that depend on these skills, such as lesion localization, understanding complex anatomical relationships, and planning surgical pathways for intraoperative guidance, will potentially benefit from this technology. In previous studies, these neurosurgical scenarios often employed in MRN evaluations serve to validate the system’s practicality and efficacy [8,11,12,22,39]. Within this study, the capabilities of the LCS-MRN system were analyzed in a diverse patient cohort of 19 human subjects, covering various pathologies, locations, and surgical objectives, including lesion resection, craniotomy planning, hemorrhage drainage, and endoscopic procedures. This analysis, conducted in a simulated clinical environment with patient imaging data, demonstrated the system’s technical performance and overall utility, confirming its effectiveness in supporting neurosurgical procedures.

The concept of accuracy in MRN systems is often variably interpreted. Still, the TRE remains a widely recognized measure for gauging navigation accuracy, applicable from initial registration stages to later phases of surgical intervention. Despite potential complications such as brain-shift-induced non-linear deformations during later surgical stages, accurate initial registration is paramount, as it fundamentally influences the accuracy and dependability of all subsequent procedural stages [26]. The TRE quantifies the discrepancy between specific points in the virtual environment and their real-world counterparts. The TRE of LCS-MRN in this study was evaluated in a three-dimensional setting. The analyses showed an average TRE of 3.0 ± 0.5 mm, ranging from 0.6 mm to 5.4 mm. This outcome aligns closely with the TRE (3.7 ± 1.7 mm) observed in the single-case pilot study of LCS-MRN [7], as well as another previous study on LCS-MRN using a fiducial-based approach showing an accuracy of 4.1 mm (interquartile range IQR 3.0–4.7 mm) [8], and is also in line with the reported TRE in conventional navigation systems applying fiducial-base registration (3.49 ± 1.09 mm) [45]. There appears to be a slight increase in accuracy with the proposed method employed in this study compared to the earlier approach, which might be related to automated approaches. However, it is important to note that these two approaches cannot be directly compared. The fiducial-based method’s accuracy was assessed using the projection error of lesions at the skin surface level as the metric for precision evaluation. While plenty of studies have examined the 3D TRE in MRN systems utilizing standalone HMDs, it is crucial to acknowledge that direct comparisons of TRE across these studies are not feasible, given the variations in their objectives and methodologies. The differences in these underline the unique aspects and context of each approach.

In mathematical terms, the TRE at specific points within this study’s context is considered a norm, necessitating the measurement of both virtual and physical points within the same coordinate space, either entirely virtual or physical. Distinct coordinate systems for each point prevent meaningful, direct measurements due to representation, scale, and direction differences. Most previous studies measured the TRE in the physical space using tools such as calipers, directly gauging the perceived distance between virtual and physical points [8]. While straightforward, this approach risks underestimating the TRE, as virtual targets could be located underneath the patient or phantom. For deep-seated targets, alternative methods involve marking virtual point locations with physical instruments (e.g., puncture needles, bone anchors) and re-registering these on post-operative images to calculate TRE [11,22,36,39,43]. This invasive approach, however, may alter the original structure and affect reliability.

In contrast, virtual space measurements offer repeatability and non-invasiveness [46,47,48]. The previous work regarding LCS-MRN developed a virtual probe for non-invasive, direct manual acquisition of actual marker points in virtual space, measuring the distance to virtual target counterparts [7]. However, moving and placing the virtual probe is subject to human error. To mitigate this, the study’s evaluations were conducted by a single neurosurgeon (Z.Q.) with extensive software/hardware and neurosurgery expertise. The FLE was introduced as a quality control parameter to reflect the potential inaccuracy due to manual operation. Worthy of note in the study’s design is that the placement of the point set V preceded that of point set P, allowing for the early acquisition of the reference metric FLE to prevent the accumulation of errors over time from affecting these baseline data, which reflect the quality of the assessment. Results indicated that FLE did not exceed TRE in any case, suggesting the manual error was well-monitored and did not compromise the credibility of TRE measurements.

Another issue worth noting is that the TRE indicates deviations at specific points, namely markers affixed to the skin surface, and does not necessarily represent the accuracy at the target level, such as lesions, eloquent structures, or proposed pathways, which remain inaccessible at that stage [7,12,49]. The study adopted alternative non-invasive approaches to address this limitation. By computing the transformation $T_{C}^{P*}$ from the fiducials’ centroids in RICS to their observed physical counterparts, the positions of anatomical structures beneath the surface in the physical world are extrapolated in the RICS. Consequently, this allows for using a set of similarity metrics, such as DSC and HD_95_, to represent the degree of overlap between virtual and physical models from volumetric and surface perspectives. While the two metrics are commonly used and effective in the context of image segmentation [50,51], their application in assessing the accuracy of AR or MR navigation systems, this methodology aligns with previous approaches [52,53], where DSC and Hausdorff Distance were similarly employed for assessment of navigation accuracy. Moreover, the study advocates using the “Segment compare” extension module in the open-source software platform 3D Slicer to calculate the DSC and HD_95_. This approach effectively ensures convenience and repeatability in the evaluation process.

The LCS-MRN system, designed as a markerless registration framework, presents an intriguing paradox in its validation approach, which employs physical markers on a head phantom for verification purposes. Researchers collected perceived virtual and real point sets from these specific markers. It is important to note that all quantitative accuracy data in this study, both interpolative and extrapolative, are derived from measurements of these markers, both physical and virtual. Subsequently, these measurements are extrapolated to estimate the overall accuracy of the head model. This methodology may raise concerns regarding the representativeness of the extrapolated whole-head data, as it is potentially influenced by the distribution of the markers used in the study. Since the extrapolation is based on a limited set of points, the accuracy and reliability of the whole-head data might be skewed or limited by the spatial configuration, number, and placement of these markers [54]. This concern is not unique to the LCS-MRN system, but is a common challenge in registration methods, including fiducial-based approaches. Several studies have explored the impact of maker distribution and placement on accuracy and reliability in fiducial-based registration methods [45,55]. In essence, the fidelity of the extrapolated data in representing the system’s performance across the entire head is contingent on the assumption that the markers provide a comprehensive and uniform representation of the head’s geometry. However, it is worth highlighting the success of these markers in facilitating accurate registration, evidenced not only in the context of the landmark-based MRN systems [8,23], but also in other applications, such as approaches based on an intraoperative scanner that assess TRE [1,2]. This success underscores their potential to be a reasonably reliable geometric reference for the entire head. This assertion is grounded in the premise that the strategic placement of markers, validated in earlier phases, might provide a global representativeness capable of inferring accuracy across the whole head. Extrapolation could be considered a pragmatic approach to gauge the system’s performance over the entire head region. It is important to note that, apart from landmark-based measures, other types of accuracy metrics may have been employed to assess the system’s performance comprehensively, such as angle difference [56,57], as well as parameters for nonlinear biomechanical modeling, such as Hausdorff distance [58]. Additionally, artificial intelligence (AI) and machine learning (ML) could serve as innovative strategies in image analysis, especially in object detection tasks [27], to assess and enhance the accuracy of the MRN system. For instance, regarding virtual-physical correspondence, AI can detect known physical points and their virtual counterparts, automatically computing registration quality parameters for error compensation, thereby improving its precision and reliability.

The LCS-MRN registration technique precisely projects and matches laser crosshairs on surfaces. A pilot study based on simulation modeling suggested that areas with smaller radii of curvature or more drastic curvature changes (e.g., nasal and zygomatic regions) provide more spatial information compared to larger or relatively flat curvature areas (e.g., temporal and occipital regions) [7]. This enhanced spatial information facilitates easier identification and correction of LCS deployment errors by users. However, it is important to consider the practical application of these findings in a clinical setting. In clinical practice, the visibility and accessibility of different regions on the patient’s head during image acquisition vary. For instance, the occipital region is typically not visible during image acquisition, which poses challenges when relying on crosshair lines in this area for registration. In contrast, registration in the supine and lateral positions can use crosshair lines marked in the nasal, zygomatic, and temporal regions, which are more visible and accessible. The heterogeneous curvature of the human head implies that different challenges and accuracies may arise when performing LCS-MRN registration and localization in various body positions. Landmark-based and lesion-based accuracy analyses indicated larger registration errors in the prone position, as evidenced by increased TRE, FN, HD_95_, and DSC values. Although post hoc analysis of the FN did not reveal pairwise statistically significant differences after Bonferroni-correction, the significantly heightened FN without correction for multiple comparisons in the prone compared to the supine position remains noteworthy, especially considering the conservative nature of Bonferroni correction, which increases the risk of Type II errors in statistical inference. A plausible explanation for this observation is that registration in supine and lateral positions typically utilizes crosshair lines marked in the nasal and zygomatic regions, as well as the temporal region, whereas, in the prone position, registration predominantly relies on crosshair lines in the occipital and temporal regions. The latter scenario involves greater use of flatter areas, potentially increasing the likelihood of LCS deployment errors. In the lesion-based accuracy analysis, the magnitudes of DSC and HD_95_ correspond well with the findings of the landmark-based analysis. However, despite some statistically significant differences in pairwise comparisons in the subgroup analysis, these differences cannot yet be considered engineering or clinical significance due to the imbalance in characteristics between the different positioning groups and small effect sizes.

Furthermore, the results indicated a positive correlation between the lesion volume and the DSC and a significant difference in LCS performance when localizing large lesions compared to smaller ones. The influence of lesion volume on DSC is frequently reported in automatic image segmentation [50]. However, to our knowledge, this is the first time this study has identified it in the MRN domain. These findings may suggest that the system is more effective in localizing larger lesions than smaller ones. No significant correlations were found between lesion depth and the distance of the lesion from the RICS origin. Future research might require an increased sample size or enhanced effect size to explore these aspects more thoroughly.

Some limitations in the introduced LCS-MRN system and within this study must be addressed in future work. Although the LCS-MRN system has demonstrated effectiveness in the controlled environment of a phantom study, translating these results to a clinical setting introduces additional complexities. The requirement to draw marker lines on the patient’s skin surface during the scanning process for LCS-MRN registration carries several limitations and potential negative effects. Skin movement across different body positions can affect the accuracy and stability of these markers, as the skin may not be in the same position during surgery as it was during scanning. This is particularly in cases where the surgical position differs from the scanning position, such as supine versus prone, which is not unique to the LCS-MRN, but also to conventional navigation systems [59]. Moreover, obtaining accurate marker lines for surgeries in prone or 3/4 prone positions can be challenging. When patients are scanned supine, the skin area required for the prone surgery markers is not exposed to the laser used for marking and is compressed or stretched, thereby hindering the line acquisition. Thus, the situation may introduce complexities not encountered in the controlled setting of a phantom study. Additional concerns include the risk of the lines being smudged or erased during patient transfer or preparation, which could compromise the registration process. All these factors underscore the need for a more robust and reliable method of establishing a reference frame for LCS-MRN registration in a clinical setting. Regarding this, a set of studies by Perkins et al. [60,61,62], conducted in the context of breast surgery, where skin movement and geometrical distortion are prevalent, opens a new pathway for enhancing LCS-MRN systems. Their conclusive evidence that patterns printed with magnetic ink can be imaged and identified via MRI paves the way for using temporary, flexible skin adhesives printed with this specialized ink as dual-visible skin markings. These markings promise to remain discernible to the naked eye and within MR imaging, potentially significantly improving current skin marking techniques. It can be hypothesized that applying such MR-visible markers and adhesives could substantially mitigate the geometric distortions caused by skin movement in neurosurgical procedures, which are similarly subject to changes in skin topology between imaging and surgery. This would preserve the integrity of surgical navigation markers and contribute to the precision of LCS-MRN registration. Moreover, incorporating MR-visible ink into the LCS-MRN workflow could potentially enhance procedural efficiency by reducing the need for manual line drawing and realignment, thereby shortening the preparation time and decreasing the potential for human error. Thus, further research and development are necessary to integrate MR ink and grid stickers into clinical LCS-MRN systems, including validation studies to confirm their effectiveness and safety.

The current workflow of the LCS-MRN system, although based on the 3D Slicer platform, involves a degree of complexity due to the reliance on multiple extension modules. Future research will aim to develop a dedicated extension module for the LCS-MRN to streamline this process. This specialized module would integrate key functionalities to enhance system efficiency and user experience. It would encompass the compatibility processing of holograms, linear transformations, and the evaluation of reverse engineering metrics. By consolidating these processes into a user-friendly module, the system’s overall complexity can be significantly reduced, leading to a more streamlined and efficient workflow.

Building on the existing LCS-MRN system’s framework, the next development phase will focus on enhancing real-time capabilities and addressing the current limitations in neurosurgical planning. The current LCS-MRN system and its workflow may not fully support real-time neurosurgical planning as traditional standard navigation systems do, such as real-time planning via a workstation with immediate feedback to the surgeon. This limitation arises from the inability of HL-2 to independently perform complex and computationally intensive tasks like image segmentation and 3D reconstruction. These processes are currently executed on separate computers before being transferred to HMDs for holographic visualization. However, the LCS-MRN system, as reported in this study, introduces an innovative approach through specialized MR interactions, notably via virtual probe placement for digitized marking. This feature, distinct from conventional navigation systems, enables a form of indirect ’real-time’ planning. It allows surgeons to non-invasively and non-destructively mark crucial points or pathways intraoperatively, computing their spatial information, which is a unique advantage over traditional navigation systems in real-time planning scenarios. This approach is particularly practical and promising for identifying and compensating for intraoperative brain shift.

The LCS system exhibits certain limitations, evident through its visualization outcomes, aligning with findings from previous studies. The system faces tracking instability, particularly at certain viewing angles, due to its dependence on the Vuforia SDK and the HL-2’s PV camera [63]. This instability is particularly problematic in the ‘Freeze’ mode, designed to stabilize the hologram’s position. In addition, there is an observed discrepancy between the positions used in preoperative imaging and those in the surgery, such as prone versus supine, leading to issues in registration accuracy. To address these challenges, a two-fold approach could be considered. Enhancing tracking stability might involve implementing alternative or supplementary tracking methods, which could provide stability across a wider range of viewing angles and overcome the limitations of the current hardware. Solutions for optimizing registration in various surgical positions could include aligning the scanning protocol of the preoperative imaging with the actual surgical positioning or integrating intraoperative imaging [59,64]. These methods would dynamically adapt the LCS system to the surgical environment, potentially further improving the accuracy and reliability of registration.

In addition to the broader challenges identified with the current methodology, the study is subject to some unique limitations. The current study is primarily a technical validation and proof of concept for the LCS-MRN system based on simulated neurosurgical procedures. It acknowledges the limitations in fully assessing its real-world effectiveness and applicability. Acknowledging this constraint, future research will involve methodical trials with human subjects better to understand the system’s performance in clinical scenarios and aim to conduct long-term studies involving actual surgical procedures. Thus, providing more comprehensive evidence of the system’s practical utility and reliability in real-world neurosurgical settings will be the next step. Moreover, while this initial investigation expands upon previous LCS-MRN research, it remains limited by an unbalanced sample size and a limited number of testers, which has constrained the study’s ability to extensively explore user experience and training. Recognizing this limitation, future studies are planned to more comprehensively evaluate the learning curve and user experience of the LCS-MRN system, especially in comparison to conventional paradigms, aiming to determine its ease of use and training needs for surgeons, which is vital for understanding its clinical practicality. In addition, this study did not include a comparative analysis with other state-of-the-art MRN paradigms due to objective constraints. Nevertheless, the evaluation methodology outlined, along with lesion-based and landmark-based metrics, facilitates future comparisons with various systems. Future research is anticipated to conduct these comparative studies, aiming for a comprehensive assessment of MRN technologies in neurosurgery.

Despite its challenges, the LCS-MRN system is promising and warrants continued refinement. The LCS-based registration improves earlier MRN registration techniques, offering a more straightforward, easy, and user-friendly process. Its streamlined hardware and software setup, coupled with the potential for increased accuracy and flexibility during potential surgical interventions, establishes this method as a noteworthy progression in developing cost-effective, user-friendly MRN systems for neurosurgical applications, leading to enhanced surgical results even in the absence of standard commercial navigation systems.

## 5. Conclusions

This study demonstrated the viability and accuracy of an LCS-MRN system through the utilization of phantoms and holograms based on patient data (MRI/CT imaging), retrospectively, particularly focusing on cases with intracranial lesions, investigating landmark- and lesion-based accuracy metrics. The findings indicate the LCS-MRN system’s potential as a technique with minimal user dependency, cost-effectiveness, and high reliability. The system’s operational efficiency and user-friendly design offer a noticeable improvement over conventional methods. With a straightforward setup in both hardware and software aspects, the system shows promise in improving accuracy and flexibility in surgical interventions. While further enhancements in precision can be anticipated, this study establishes a solid foundation for future development in accessible, user-friendly MRN technologies. Such advancements are pivotal in potentially elevating the quality of surgical procedures. Ongoing research is expected to refine further and validate these attributes in a wider range of clinical applications.

## Figures and Tables

**Figure 1 sensors-24-00896-f001:**
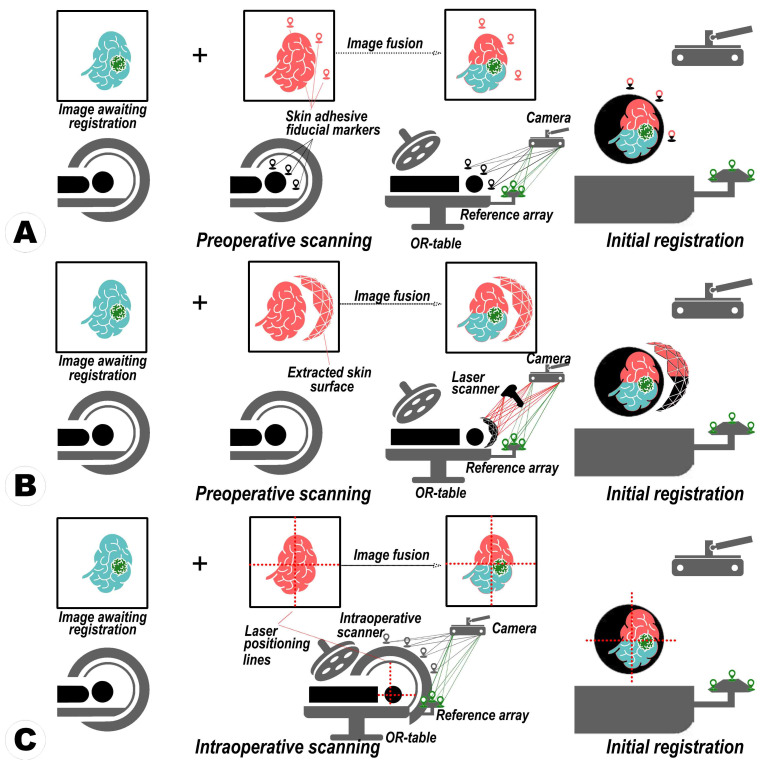
An illustration of general paradigms in conventional navigation systems. (**A**) Fiducial-based registration employs identifiable external markers placed on the patient’s skin. These are reference points to align preoperative images with the patient’s physical space during surgery. (**B**) Surface-based registration utilizes the contours of the patient’s exposed surfaces to create a spatial map that aligns preoperative images with the patient’s anatomy in the operating room. (**C**) Automatic registration effectively aligns the captured three-dimensional (3D) volumetric data (reference image) with its origin of capture, utilizing scanner tracking mechanisms such as those in intraoperative scanners to guarantee precise alignment for navigation purposes.

**Figure 2 sensors-24-00896-f002:**
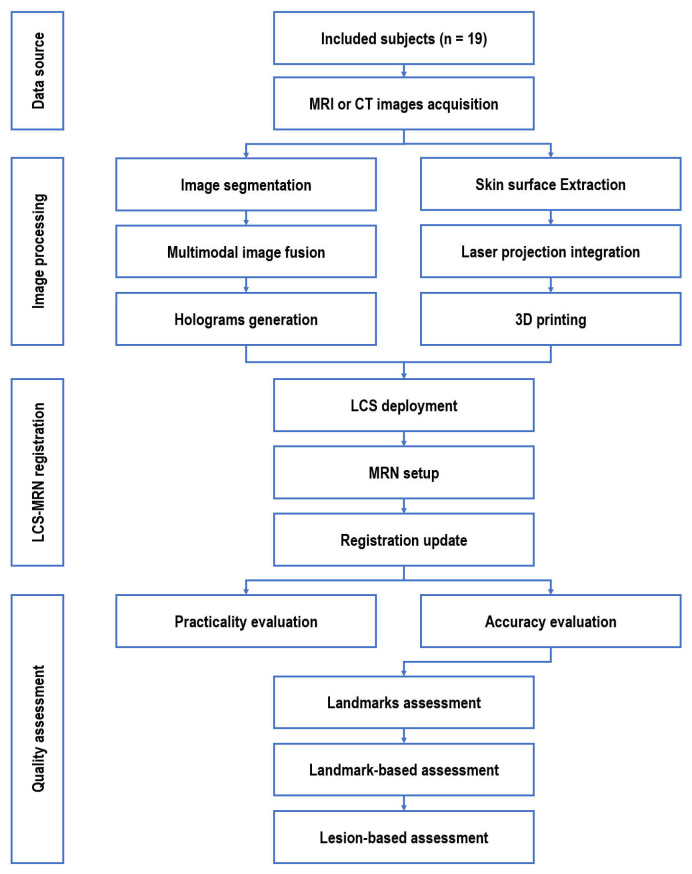
Schematic of the study framework. LCS = laser crosshair simulator; MRN = mixed reality navigation.

**Figure 3 sensors-24-00896-f003:**
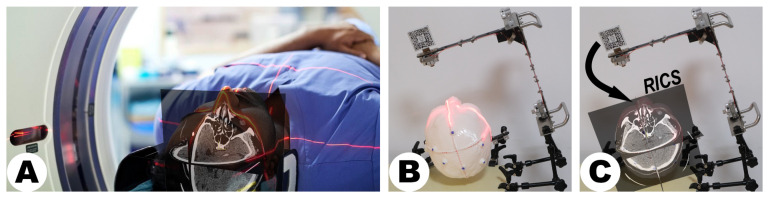
Functionality of the laser crosshair simulator (LCS). The LCS replicates a scanner’s laser crosshairs, projecting dual laser lines onto a patient’s head as seen in CT or MRI environments (**A**,**B**). Its primary function is to facilitate the translation of scanning parameters across various spatial and temporal settings to establish the position for the reference image coordinate system (**C**). The black curved arrow links the tracking and virtual environments via the MR interface. Detection and recognition of the target images by the HMD prompt the initialization of the virtual space, anchoring its origin at the predetermined position. ((**A**) provided by Alamy, has undergone minor modifications by the authors to suit the specific context of this work. Alamy has granted copyright authorization for the use and adaptation of the image). RICS = reference image coordinate system.

**Figure 4 sensors-24-00896-f004:**
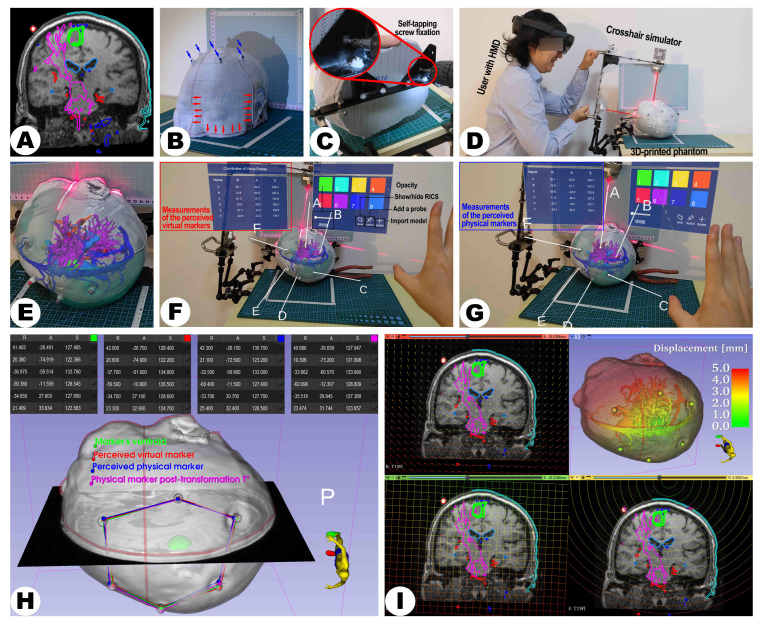
The figure illustrates the technical validation process of the LCS-MRN system using the data of a 73-year-old female patient with a right parietal metastasis close to eloquent motor structures. (**A**) displays T1-weighted MRI data with key anatomical structures (green—lesion, bright red—markers, cyan—scalp quadrants, deep orange—arteries, dark blue—venous sinuses, light blue—ventricles, and magenta—pyramidal tract), markers, and laser positioning lines highlighted for 3D reconstruction. (**B**) shows the 3D-printed head phantom with integrated markers (blue arrows) and laser positioning lines (red arrows). (**C**) depicts the phantom affixed to a head clamp, indicating fixation points (red circles). (**D**) illustrates the deployment of the LCS and alignment of the laser crosshairs. (**E**) presents the interaction with holograms through the MR platform. (**F**,**G**) display the virtual probe positioning on virtual and physical markers, respectively. The visualization of the registration process and the subsequent data analysis, including the calculation of target registration error (TRE) and other metrics, is visually represented in (**H**,**I**). Detailed descriptions of each step, particularly the methodologies for measuring points and data processing, are discussed in the main text.

**Figure 5 sensors-24-00896-f005:**
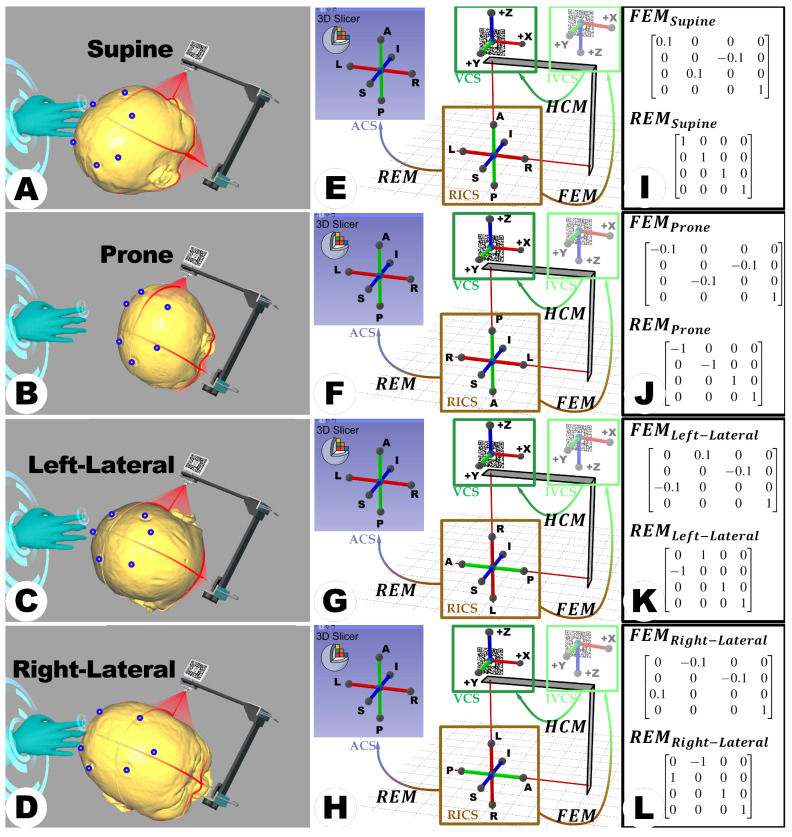
Compatibility processing of holograms for varying surgical positions. Panels (**A**–**D**) display the head phantom in supine, prone, left-lateral, and right-lateral positions, respectively, highlighting the LCS-MRN system setup. Panels (**E**–**H**) illustrate the different orientations of the RICS (brown) in comparison to the intermediate VCS (IVCS) (light green), the virtual coordinate system (VCS) (dark green), and the analysis coordinate system (ACS) (light blue) within the 3D Slicer software (Version 5.1.0) for each position, as well as their transformations (indicated by gradient color arrows): the Forward Engineering Matrix (FEM), Reverse Engineering Matrix (REM), and Handedness Conversion Matrix (HCM). Panels (**I**–**L**), representing the FEM and REM, are customized for each surgical position to ensure accurate registration and analysis of the holograms within the LCS-MRN system. This process guarantees the precise display and visualization of MR content across different surgical scenarios.

**Figure 6 sensors-24-00896-f006:**
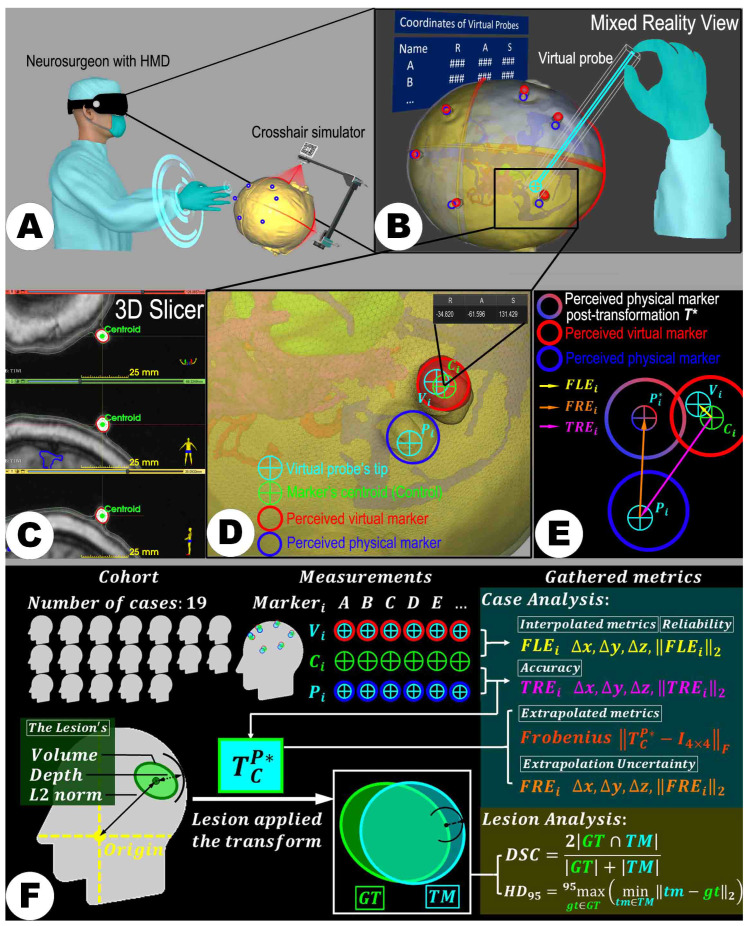
An illustrative scheme demonstrates the quality assessment procedure. After registration of a set of holograms and the physical space (**A**) using the LCS, achieving theMRN registration, the tip of the virtual probe is positioned at the perceived centers of the virtual markers (red circles) and the physical markers (blue circles), respectively, and their 3D coordinates (automatically transformed into the reference image coordinate system (RICS)) were immediately reported and displayed (where ### represents the numerical values of R, A, or S) on the MR platform (**B**). Next, the coordinates reported underwent REM transformation. They were subtracted from the coordinates of the marker’s centroid, which were annotated in advance in 3D Slicer (**C**,**D**), resulting in two displacement vectors, i.e., fiducial localization error **FLE** and target registration error **TRE** (**E**). Moreover, the optimal rigid transformation $T_{C}^{P}$ was calculated, ensuring the calculation of a set of extrapolative metrics, such as the fiducial registration error **FRE** and Frobenius norm **FN**. When $T_{C}^{P}$ is applied to the original segmentation of the lesion (as the ground truth, **GT**), the transformed model (**TM**) is obtained, thereby allowing the calculation of the Sørensen–Dice similarity coefficient (DSC) and the 95% Hausdorff distance (HD_95_), focusing on the LCS-MRN accuracy regarding the lesion volume and shape (**F**).

**Figure 7 sensors-24-00896-f007:**
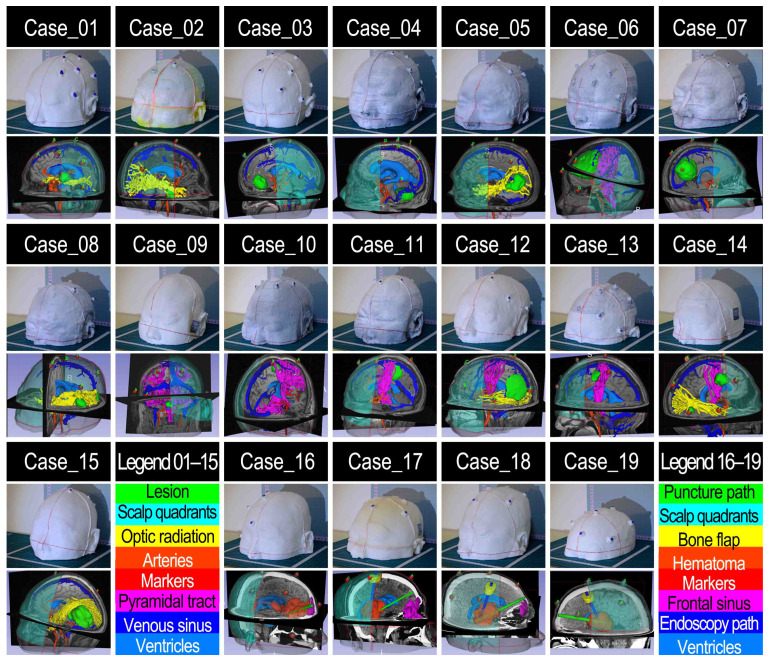
3D-printed phantoms (rows 1, 3, 5) and corresponding holograms (rows 2, 4, 6) were generated from imaging data in all included 19 cases.

**Figure 8 sensors-24-00896-f008:**
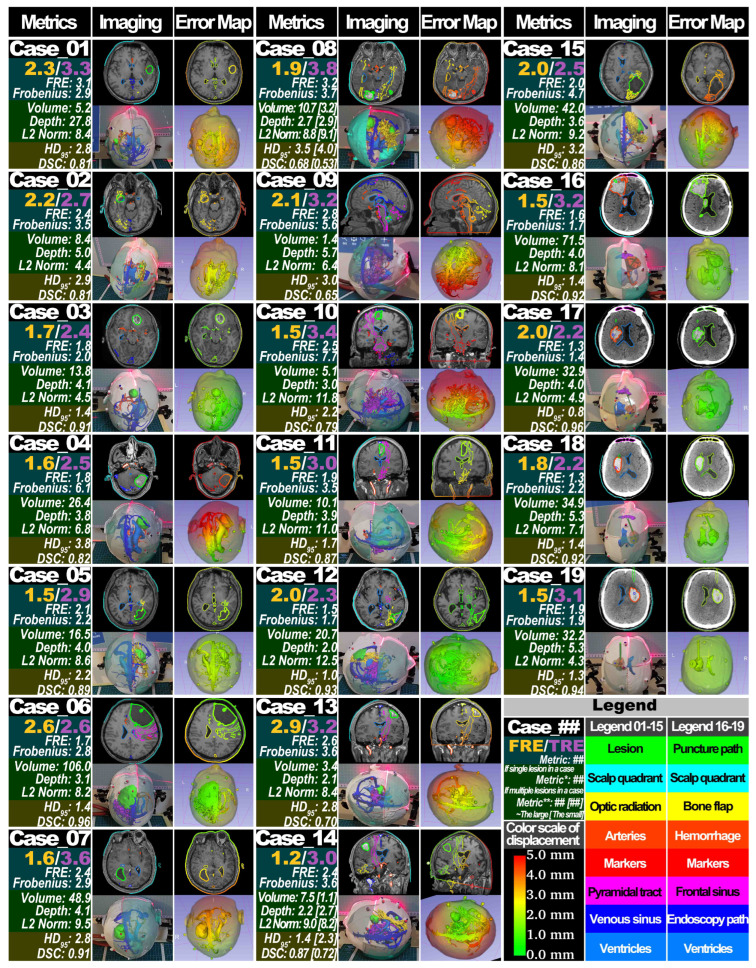
Visual outcomes and corresponding metrics for all 19 cases utilizing the LCS-MRN system. Each case features a metrics list displayed on the left column; holograms derived from segmentation results are exhibited in the central column, with two-dimensional (2D) contour representations on the top and 3D models displayed within an MR setting on the bottom. On the right column of a case display, extrapolated correspondence errors of the registered holograms are illustrated using pseudocolor scale maps. For comparative ease, identical planes from 2D images and consistent perspectives within the 3D space are utilized. Within this figure, the legend positioned at the lower right corner elucidates the scale map legend, hologram labeling, and the significance of the metrics list. In the legend, the symbol “##” denotes a numerical value. A single asterisk “*” signifies one lesion, and a double asterisk “**” two in a case. It is important to note that for cases with two lesions (Case_08 and Case_14), the metrics reported outside the brackets pertain to the larger lesion, while values within the brackets correspond to the smaller lesion.

**Figure 9 sensors-24-00896-f009:**
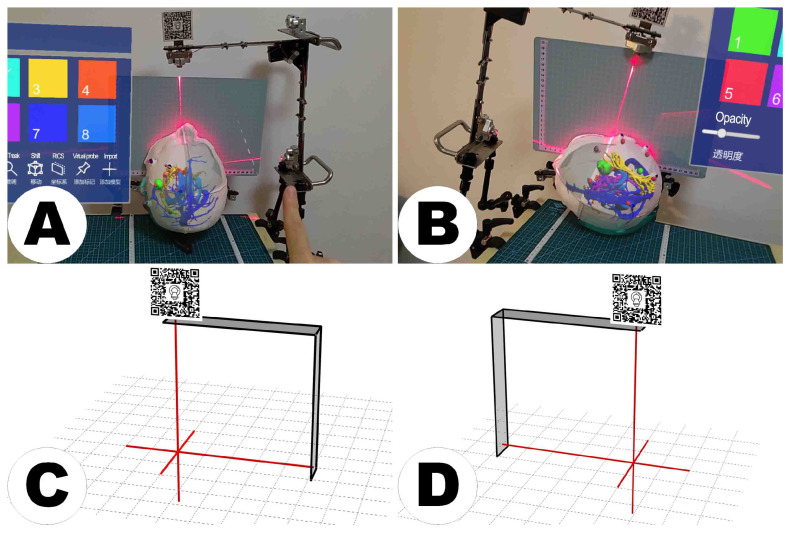
Dual-sided deployment of LCS. Panels (**A**,**B**) exhibit the LCS’s MR interface with Vuforia image targets attached on both sides, captured in a simulated operative context with the LCS positioned on different sides of the phantom head. The images clearly demonstrate successful registration and visualization of the holograms, indicating reliable recognition and tracking by the LCS-MRN system. Panels (**C**,**D**) offer schematic diagrams, emphasizing the LCS’s placement versatility around the user, accommodating both left-sided and right-sided arrangements.

**Figure 10 sensors-24-00896-f010:**
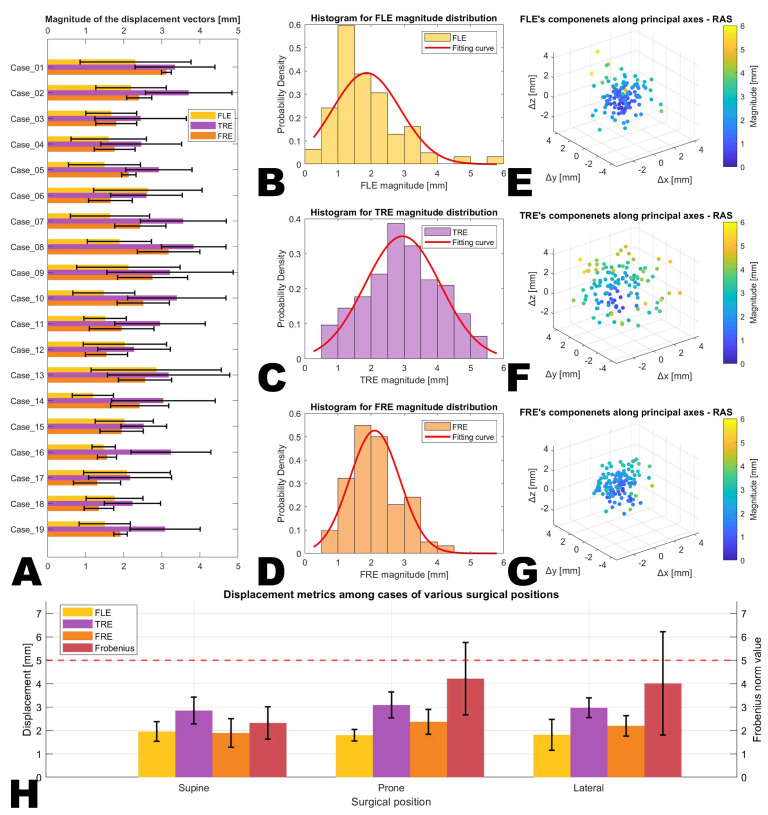
Landmark-based accuracy results. The bar chart displays the magnitude of displacement vectors (FLE, TRE, and FRE) for each case (**A**). Histograms show the distribution of the magnitudes of each displacement vector across all measurements with a fitted normal curve (**B**–**D**). 3D scatter plots are based on the components of each displacement vector along the three primary axes of the reference image coordinate system, with color gradients representing vector magnitude. (**E**–**G**). Clustered bar charts illustrate comparisons of displacement metrics (FLE, TRE, FRE, and FN) grouped by surgical position (**H**). The whiskers in (**A**,**H**) on the plot extend to the smallest and largest data points that fall within a distance of 1.5 times the interquartile range (IQR) from the lower (Q1) and upper (Q3) quartiles, respectively.

**Table 1 sensors-24-00896-t001:** Demographic data.

Case	Sex [M/F]	Age [Years]	Pathology	Localization	Surgical Position
01	M	49	Metastasis	Left temporal	Supine
02	M	58	Diffused astrocytoma	Right temporal	Supine
03	M	8	Cavernous malformation	Left frontal	Supine
04	M	62	Meningioma	Left cerebellar	Prone
05	M	41	Diffuse large B-cell lymphoma	Left occipital	Prone
06	F	27	Meningioma	Left frontal	Supine
07	M	51	Metastasis	Right occipital	Prone
08	F	66	Metastasis	Bilateral occipital	Prone
09	F	37	Aneurysm	Fourth ventricular	Prone
10	F	73	Metastasis	Right parietal	Left-lateral
11	F	54	Meningioma	Left parietal	Left-lateral
12	M	79	Metastasis	Left occipital	Right-lateral
13	F	41	High grade glioma	Left parietal	Right-lateral
14	M	74	Metastasis	Right frontal and occipital	Left-lateral
15	F	58	Metastasis	Left occipital	Prone
16	M	84	Hematoma	Right frontal	Supine
17	M	63	Hematoma	Right basal ganglia	Supine
18	M	57	Hematoma	Right basal ganglia	Supine
19	M	51	Hematoma	Left basal ganglia	Supine

**Table 2 sensors-24-00896-t002:** Case-specific reference images and individual hologram content.

Case	Reference Image	L ^1^	A ^2^	VS ^3^	V ^4^	FS ^5^	OR ^6^	PT ^7^	PP ^8^	EP ^9^	BF ^10^	M ^11^	SQ ^12^
01	T1	✔	✔	✔	✔	-	✔	-	-	-	-	7 *	✔
02	T1	✔	✔	✔	✔	-	✔	-	-	-	-	7	✔
03	T1	✔	✔	✔	✔	-	-	-	-	-	-	7	✔
04	T1	✔	✔	✔	✔	-	-	-	-	-	-	7	✔
05	T1	✔	✔	✔	✔	-	✔	-	-	-	-	7	✔
06	T1	✔	✔	✔	✔	-	-	✔	-	-	-	7	✔
07	T1	✔	✔	✔	✔	-	-	-	-	-	-	7	✔
08	T1	✔ (2)	✔	✔	✔	-	✔	-	-	-	-	7	✔
09	T1	✔	✔	✔	✔	-	-	✔	-	-	-	6	✔
10	T1	✔	✔	✔	✔	-	-	✔	-	-	-	6	✔
11	T1	✔	✔	✔	✔	-	-	✔	-	-	-	6	✔
12	T1	✔	✔	✔	✔	-	✔	✔	-	-	-	6	✔
13	T1	✔	✔	✔	✔	-	-	✔	-	-	-	6	✔
14	T1	✔ (2)	✔	✔	✔	-	✔	✔	-	-	-	6	✔
15	T1	✔	✔	✔	✔	-	✔	-	-	-	-	6	✔
16	CT	✔	-	-	✔	✔	-	-	✔	-	-	7	✔
17	CT	✔	-	-	✔	✔	-	-	✔	✔	✔	6	✔
18	CT	✔	-	-	✔	✔	-	-	✔	✔	✔	7	✔
19	CT	✔	-	-	✔	-	-	-	✔	✔	✔	7	✔
Overall	Count	21	15	15	19	3	7	7	4	3	3	124	19

^1^ Lesion(s); ^2^ arteries; ^3^ venous sinus; ^4^ ventricle; ^5^ frontal sinus; ^6^ optic radiation; ^7^ pyramidal tract; ^8^ puncture path; ^9^ endoscopic path; ^10^ bone flap; ^11^ fiducial markers; ^12^ scalp quadrants. “✔” = segmented; “-” = not segmented; * number of available markers; “(2)” = number of lesions.

**Table 3 sensors-24-00896-t003:** Landmark-based accuracy metrics among all surgical positions.

Characteristic	FLE [mm]	TRE [mm]	FRE [mm]	FN ^†^
Overall (n = 19)	1.9 ± 0.4	3.0 ± 0.5	2.1 ± 0.6	3.4 ± 1.7
Supine (n = 8)	2.0 ± 0.4	2.9 ± 0.6	1.9 ± 0.6	2.3 ± 0.7
Prone (n = 6)	1.8 ± 0.2	3.1 ± 0.6	2.4 ± 0.5	4.2 ± 1.5
Lateral (n = 5)	1.8 ± 0.7	3.0 ± 0.4	2.2 ± 0.4	4.0 ± 2.2
*p*-value *	0.628	0.745	0.154	0.034

^†^ Frobenius norm. * Kruskal–Wallis test.

**Table 4 sensors-24-00896-t004:** Lesion-based accuracy metrics among all surgical positions.

Characteristic	Volume [cm^3^]	Depth [cm]	L2 Norm [cm]	DSC	HD_95_ [mm]
Overall lesions (n = 21)	23.9 ± 26.3	3.8 ± 1.5	8.0 ± 2.3	0.83 ± 0.12	2.3 ± 1.0
Supine lesions (n = 8)	38.1 ± 34.6	4.7 ± 1.7	6.2 ± 1.9	0.90 ± 0.06	1.7 ± 0.8
Prone lesions (n = 7)	21.3 ± 18.6	4.0 ± 1.0	8.4 ± 1.2	0.76 ± 0.15	3.2 ± 0.6
Lateral lesions (n = 6)	8.0 ± 7.0	2.6 ± 0.7	10.1 ± 1.8	0.81 ± 0.09	1.9 ± 0.7
*p*-value *	0.069	0.012	0.005	0.005	0.042

* Kruskal–Wallis test.

**Table 5 sensors-24-00896-t005:** Correlation analysis between lesion-based accuracy metrics and lesion characteristics.

Spearman’s Rank Correlation	*rho* Value	*p*-Value
DSC—volume [cm^3^]	0.813	<0.001
DSC—depth [cm]	0.354	0.115
DSC—L2 norm [cm]	−0.201	0.380
HD_95_ [mm]—volume [cm^3^]	−0.377	0.092
HD_95_ [mm]—depth [cm]	−0.196	0.395
HD_95_ [mm]—L2 norm [cm]	0.159	0.490

**Table 6 sensors-24-00896-t006:** Subgroup accuracy analysis of radiological characteristics.

		HD_95_ [mm]			DSC	
Characteristic	>Median	≤Median	*p*-Value *	>Median	≤Median	*p*-Value *
Volume	1.9 ± 1.0	2.6 ± 0.8	0.112	0.91 ± 0.04	0.76 ± 0.11	<0.001
Depth	1.9 ± 0.8	2.6 ± 1.03	0.148	0.88 ± 0.09	0.79 ± 0.12	0.084
L2 norm	2.5 ± 1.0	2.1 ± 1.0	0.306	0.80 ± 0.13	0.86 ± 0.10	0.245

* Mann–Whitney U-test.

**Table 7 sensors-24-00896-t007:** Comparative analysis of registration techniques in state-of-the-art MRN paradigms.

	Manual Alignment	Fiducial-Based Registration	Markerless Registration	LCS ^†^-Based Registration
Advantages	Simplicity and portability; MR-HMD self-tracking; user perspective adaptability	Speed advantage; real-time AR marker detection; rigid tissue anchoring	Contact hazard elimination; automated surface data acquisition	Simplified process; low user-dependency; non-reliance on calibration objects; global averaging; simple assembly; cost-effectiveness
Limitations	Time consumption; lack of real-time support; spatial map coarseness; static nature; adjustment needs for spatial shifts	Complete user dependency; marker shift errors; environmental susceptibility; prone position inaccuracy	robustness challenges; noise sensitivity; high computational demand; prone position non-availability	Marker line drawing difficulties; tracking angle and distance susceptibility
Cited works	Li et al., 2018 [11]; Li et al., 2023 [22]; Gibby et al., 2019 [36]; McJunkin et al., 2019 [37]; Marrone et al., 2024 [38]	Qi et al., 2021 [8]; Qi et al., 2022 [23]; Gibby et al., 2021 [39]; Zhou et al., 2023 [40]; Eom et al., 2022 [41]; Akuluaskas et al., 2023 [42]	Liebmann et al., 2019 [43]; Pepe et al., 2019 [32]; von Haxthausen et al., 2021 [44]	Qi et al., 2023 [7]

^†^ Laser crosshair simulator.

## Data Availability

The data presented in this study are available upon reasonable request from the corresponding authors.

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
