# Peer review of "The Feasibility and Accuracy of Holographic Navigation with Laser Crosshair Simulator Registration on a Mixed-Reality Display"

_sensors, 2024, doi:10.3390/s24030896_

Round 1

Reviewer 1 Report

Comments and Suggestions for Authors

Thank you for the opportunity to review the manuscript titled 'Feasibility and Accuracy of Holographic Navigation with Laser Crosshair Simulator Registration on a Mixed-Reality Display.' Your work on a cost-effective mixed-reality navigation system using a laser crosshair simulator integrated with Microsoft's HoloLens-2 is an innovative approach in neurosurgical navigation. The study, utilizing 3D-printed head phantoms from patient data to assess system accuracy, is commendable. However, there are areas in the manuscript needing further revision and clarification. Below are some key suggestions for refinement before publication.

Strengths of the Paper:

1. Innovation and Relevance: The study addresses a significant issue in neurosurgery – the need for more cost-effective and user-friendly navigation systems. The integration of MRN with LCS in a clinical setting is a novel approach that can potentially transform surgical navigation techniques.

2. Methodological Rigor: The study's methodology is thorough. It involves retrospective data collection from patients diagnosed with intracranial lesions, ensuring high-resolution and clear imaging data. The use of life-sized 3D head phantoms for simulation is a robust approach for assessing the system's feasibility and accuracy.

3. Comprehensive Analysis and Validation: The paper conducts a detailed analysis of the LCS-MRN system's performance, including practicality assessment, landmark-based evaluation, and lesion-based evaluation. The use of various metrics like TRE, FLE, FRE, and FN for accuracy assessment demonstrates a comprehensive approach to validation.

Areas for Improvement:

1. Clarity and Detail in Reporting Results: While the paper presents detailed methodologies, the results section could benefit from more clarity and depth in reporting and discussing the findings. This would help readers better understand the implications of the study’s findings.

2. Comparative Analysis: The study could be strengthened by including a comparative analysis with existing conventional systems. This would provide a clearer picture of the LCS-MRN system's advantages and limitations relative to current standards.

3. Broader Application and Scalability: The paper could discuss the potential scalability of this technology and its applicability in various clinical settings. Addressing how this system could be adapted or scaled for different surgical procedures or environments would add value to the study.

4. Long-term Clinical Evaluation: While the study effectively uses phantoms for evaluation, long-term studies involving actual surgical procedures would provide more robust evidence of the system's practical utility and reliability in real-world scenarios.

5. User Experience and Training: More information on the learning curve and user experience with the LCS-MRN system, especially in comparison to traditional systems, would be beneficial. This would help assess the system's ease of use and the training required for surgeons.

6. Technical Specifications and Limitations: The paper could elaborate more on the technical specifications and limitations of the LCS-MRN system. Understanding its technical constraints, compatibility with different surgical tools, and any potential limitations in its application would be crucial for its practical implementation.

Overall, the paper is a valuable contribution to the field of neurosurgery and medical navigation technology. With some enhancements, especially in the areas of comparative analysis, long-term clinical evaluation, and detailed discussion of results, the paper could provide a  comprehensive insight into the feasibility and practicality of the LCS-MRN system. 

Moreover, I commend the authors for their exceptional work and innovative contributions to the field, and I look forward to seeing the impact of this study in advancing neurosurgical navigation technologies.

Reviewer 2 Report

Comments and Suggestions for Authors

Dear Authors, I read your article with interest. The topic is extremely complex but  the article is well written and clear. I have a question:

- Which clinical cases might beneficiate more from this technologies? 

- Do you think artificial intelligence might be employed to overcome any of the limitations described (e.g. verify the correspondence between virtual model coming from radiological images and the real surgical field? 

I would include these two points in the discussion.

Reviewer 3 Report

Comments and Suggestions for Authors

This paper proposes a new automatic registration method for neurosurgical navigation systems. The proposed method utilizes both coplanar laser emitters and a recognizable target pattern. Experimental results show that the proposed method can achieve excellent accuracy for neurosurgical planning. The paper is generally well written. The proposed method is original and may have important applications in clinical practices. However, the following issues need to be addressed before the paper can be accepted for publication.

1. The major contributions of the paper should be clearly summarized in introduction.

2. In Section 2, a clear description of the problem that the paper intends to solve should be presented before Section 2.1.

3. In Section 2.3, a better explanation on the matrices shown in Figure 4 should be provided in the text. For example, why they are needed for registration? How are they employed to complete the registration task? etc.

4. In Section 3, is it possible to compare the accuracy of the proposed approach with that of several other state-of-the-art methods? Also, is the proposed method capable of real-time neurosurgical planning? 

Round 2

Reviewer 3 Report

Comments and Suggestions for Authors

The paper has been carefully revised to address all issues. I have no other concerns on the revised paper and recommend its acceptance.